# ARITHMETIC WITH LANGUAGE MODELS: FROM MEMORIZATION TO COMPUTATION

## ABSTRACT

A better understanding of the emergent computation and problem-solving capabilities of recent large language models is of paramount importance to further improve them and broaden their applicability. This work investigates how a language model, trained to predict the next token, can perform arithmetic computations generalizing beyond training data. Binary addition and multiplication constitute a good testbed for this purpose, since they require a very small vocabulary and exhibit relevant input/output discontinuities making smooth input interpolation ineffective for novel data. We successfully trained a light language model to learn these tasks and ran a number of experiments to investigate the extrapolation capabilities and internal information processing. Our findings support the hypothesis that the language model works as an Encoding-Regression-Decoding machine where the computation takes place in the value space once the input token representation is mapped to an appropriate internal representation.

## 1 INTRODUCTION

Large Language Models (LLMs) based on Transformer architecture (Vaswani et al., 2017) have recently demonstrated surprising problem-solving capabilities that require logic reasoning, advanced information processing and common sense (Bubeck et al., 2023; Wei et al., 2022b;a). Their huge storage capacity combined with a massive training on terabytes of heterogeneous data could suggest that the memorization of an enormous amount of knowledge is enough to perform well on similar test data. However, validations on carefully selected Out-of-Distribution (OoD) data proved their reasoning capabilities on novel examples requiring non-trivial generalizations. Unfortunately, the depth and width of such models is so high that decoding and understanding the internal information processing is very challenging.

Focusing on arithmetic calculations, some studies (Yuan et al., 2023) demonstrate that recent LLMs (such as GPT-4) can perform additions and multiplications with long-digit operands, for which the number of variants is so high to exclude the exhaustive memorization of the training set. Nevertheless, the computational approach put in place by the LLMs, as well as the interpolation/extrapolation capabilities remain unexplained.

In this work we design some controlled experiments, consisting of simple computation tasks such as binary addition and multiplication, and solve them with a Language Model (LM) based on Transformer architecture (Vaswani et al., 2017). In spite of their simplicity, these tasks cannot be solved by pure memorization or smooth interpolation and investigating how an LM learn them can improve our understanding of the underlying mechanisms. In particular, using a tiny vocabulary of just 5 tokens and a small training set allows to operate with a light (non-pretrained) LM and use interpretability techniques to investigate internal information processing.

Other studies addressed the ability of LLMs to perform arithmetic computation and train small LMs to learn these tasks from scratch (see related works in Section 2). However, our aim is different: we are not interested in finding the best LM architecture and setup to maximize accuracy on arithmetic operations, but we look for a simple architecture and setup that allow to effectively solve the task in order to be able to investigate the underlying computational approach. The main novelty and contribution of this work are the formalization of the hypothesis that our LM works as an Encoding-Regression-Decoding machine and the design of a number of experiments to support and validate this hypothesis.

After presentation of related works in Section 2, in Section 3 we introduce the experimental testbed and the architecture of the LM used. Section 4 presents the results achieved and introduces control experiments and elaborations to shed light on the computation approach used to solve the tasks. In Section 5 an ablation study is presented and, finally, in Section 6 we include a final discussion and draw some conclusions.

## 2 RELATED WORKS

### 2.1 LM AND LLM CAPABILITIES ON ARITHMETIC TASKS

In Yuan et al. (2023) recent LLMs have been benchmarked in arithmetic tasks, including long-digits sum and multiplication, showing that LLMs such as ChatGPT and GPT-4 can perform reasonably well on these tasks even with no specific tuning. The accuracy of smaller models is markedly lower, and in general they are not able to work with long operands and generalize to OdD data.

Nogueira et al. (2021) tuned a T5-based pre-trained LM on additions and subtractions, and argued that tokenization and input representation are critical to achieve good accuracy. In particular, in their experiments character-based tokenization works better than sub-word tokenization, and making explicit the digit position in the input string (i.e., inserting after each digit a marker to denote its position in the sequence) generally leads to better accuracy. They also trained a vanilla non-pretrained LM on smaller numbers and found that classical sinusoidal-based positional embedding does not perform well, so they proposed a tailored position-wise masked embedding. Their paper contains other interesting finding such as the impact of the digit order (plain or reverse) and the size of the training set.

Muffo et al. (2023) tuned pre-trained GPT-2 models on 5-digit additions and 2-digit multiplications. They also found that making explicit the digit position in the input sequence helps to improve accuracy. While good accuracy is reported for addition, the tuned models struggle to learn multiplication even on two-digit operands.

Lee et al. (2023) train small LMs to learn arithmetic tasks, mainly focusing on addition, but also experimenting with subtraction, multiplication, sine and square root. The authors carefully ablate different aspects of the training data to isolate the factors that contribute to the appearance of arithmetic capabilities. In particular, they study the impact of the input order (plain or reverse) and the utility of providing intermediate information about the decomposition of the task in steps to promote Chain of Thought (CoT) (Wei et al., 2022b) reasoning. Some results and findings included in Lee et al. (2023) will be further discussed throughout this paper.

All the above works provide useful contributions to understand the capabilities and limitations of large and small LMs to deal with arithmetic tasks, but none of them focus on the computational approach used to solve them, which is the main purpose of the present work.

### 2.2 INTERPRETABILITY TECHNIQUES

A large number of techniques can be used to investigate the internal working mode of deep neural networks, including transformers and LMs: see Räuker et al. (2023) for a recent survey. Weights, single neurons, subnetworks/circuits, and activations can be the target of *intrinsic* approaches (implemented during training) or *post-hoc* approaches (implemented after training).

Probing is a common technique used to investigate the representations learned by pre-trained LMs: it typically involves training a simple model (denoted as *probe*) on top of the LM embeddings to predict a given property (Belinkov, 2022). Moreover, structural probing can be used to check whether internal representations encode discrete structures such as syntax trees (Hewitt & Manning, 2019), (White et al., 2021). However, a certain criticism emerged on probing analyses which is believed to disconnect the probing task from the original one and/or to reveal correlations instead of causations. Therefore, instead of focusing on the presence of information on internal encoding, some researchers proposed to check whether the removal of some knowledge from embeddings negatively influences the model ability to perform a task (Elazar et al., 2021), (Lasri et al., 2022). Mechanistic interpretability is still more ambitious, since it is aimed at reverse engineering the algorithm that a model uses to solve a task and map it to neural circuits (Elhage et al., 2021).

In this work we use a mix of intrinsic and post-hoc interpretability techniques: in particular through the experiments we manipulate the training set, change the input representation and the architecture components, and perform correlations analyses of embeddings.

### 2.3 MECHANISTIC INTERPRETABILITY OF ARITHMETIC REASONING WITH LMS

Stolfo et al. (2023) introduced a causal mediation analysis to point out the LM components (e.g., attention heads, Multi-Layer Perceptrons - MLPs) involved in the information processing of simple arithmetic operations, focusing on the flow of numerical information throughout the model layers/columns. The main outcome of this study is that the model: (i) processes the representation of numbers and operators with the first layers; (ii) information is then conveyed (by attention heads) to the last part of the sequence (i.e., output column), where (iii) it is numerically processed by late MLPs.

Nanda et al. (2023) carefully studied the algorithmic approach put in place by a small Transformer to implement modular addition of small numbers. They discovered that the internal algorithmic implementation is based on discrete Fourier transforms and trigonometric identities to convert addition to rotation on a circle. While the outcomes are somewhat surprising, here the term algorithm must be taken with care: even if the experiments prove that internal processing well approximate given equations, the approach is a numerical approximation (based on weight encoded values) that does not generalize to different moduli (as a symbolic implementation of the equations could do).

Both these studies adopted a simplified setting where numbers are presented as single token, and the output is expected at the last position of the sequence. So the models are not operated in autoregressive manner and the multi-token encoding/decoding stages are simplified. In Section 6 we discuss how the above findings are compatible with our findings.

## 3 EXPERIMENT DESIGN

### 3.1 THE TASKS

We focused on two simple computation tasks: binary addition and binary multiplication. Using binary encoding allows keeping the vocabulary very compact, since we need to encode only the symbols '0', '1' and few other tokens. The selected tasks have other nice properties such as computing input similarities by Hamming distance and easily generating all combinations. Of course, a classical artificial neural network can be trained to learn to sum and multiply two integers or floating-point numbers, but adding/multiplying strings of tokens with an LM is trickier.

More formally, given two integers $A$, $B$ (both in the range [0,127]) our input sequence (or prompt) is a 14-token string taking the form:

$$a_0a_1a_2a_3a_4a_5a_6 \langle op \rangle b_0b_1b_2b_3b_4b_5b_6$$

where $a_i, b_i \in \{\text{'0', '1'}\}$ are the symbols corresponding to bits in the $i$-th position in the binary representation of $A$ and $B$ respectively, and $\langle op \rangle$ can be either '+' or '×'.

The expected output string (or input completion) is:

$$R = r_0r_1...r_m$$

$$r_i = \begin{cases} i_{th} & \text{bit in the binary representation of} & A + B, i = 0...7 & \text{if} & \langle op \rangle = \text{'+'} \\ i_{th} & \text{bit in the binary representation of} & A \times B, i = 0...13 & \text{if} & \langle op \rangle = \text{'×'} \end{cases}$$

It is worth noting that:

- we are using a fixed-length input/output representation (with zero padding for unused most significant bits) to make the digit positions more explicit.
- in both the input and output the Least Significant Bits (LSBs) are provided before the Most Significant Bits (MSBs) (a.k.a., reverse or little-endian order) since this was supposed to

simplify the model learning[1]. As discusses in Appendix C this assumption leads to a much faster training.

If we consider the sequence-to-sequence mapping underlying the proposed tasks we note that even in a simple binary addition a slight change in the input (i.e., a single bit) can produce a relevant change in the output because of the carries propagation (as shown explicitly in Appendix A).

## 3.2 THE ARCHITECTURE

A non-pretrained encoder-decoder Transformer based on the original architecture introduced by Vaswani et al. (2017) was used as LM. Table 1 reports the model setup and parametrization. The small vocabulary used allows to keep the model small (just 701K learnable parameters) and trainable from scratch with a limited number of examples.

Table 1: Details of the LM model used in our experiments. The total number of learnable parameters is just 701K, that is several orders of magnitudes smaller than recent billion-parameters LLMs.

| | |
|---|---|
| vocabulary size | 5 |
| vocabulary | 0: unused, 1: \<start\>, 2: '+' or '×', 3: '0', 4: '1' |
| token embedding | learned |
| positional encoding | fixed (sinusoidal) |
| $d_{model}$ | 64 |
| $d_{ff}$ | $d_{model} \times 4$ |
| num_heads $h$ | 8 |
| encoder layers | 6 |
| decoder layers | 6 |
| dropout | 0.1 |
| learnable parameters | 701K |

The LM was trained to learn separately the addition/multiplication tasks. For both problems, we exhaustively generated all the $2^{14} = 16384$ input/output combinations, which were then randomly split in training ($3/4 \rightarrow 12288$) and validation ($1/4 \rightarrow 4096$).

An additional control experiment was run where the input sequences are the same of the addition experiment but the output completion was randomly generated (with the same length of the addition, i.e., 8 tokens). In this case, the lack of any dependencies between input and output makes it impossible to learn an algorithmic approach (or smooth mapping) to solve the problem and the only strategy to learn the training set is memorizing all the sequences.

When the trained LM is used in inference mode, we always pick the most probable token from the logit outputs (i.e., greedy decoding). Two metrics can be used to denote the LM accuracy: *token accuracy* refers to the probability of generating the next token correctly, while *sequence accuracy* refers to the probability of generating the whole output string correctly in autoregressive mode (i.e., generating one token at a time and appending it at the current prompt).

All the experiments included in this paper can be easily reproduced by running the code available at: (to be disclosed upon acceptance).

## 4 RESULTS AND DISCUSSION

### 4.1 LEARNING ADDITION AND MULTIPLICATION

Figure 1 shows that our simple LM is able to learn addition in less than 50 epochs, and multiplication in about 250 epochs [2]. As expected multiplication is more complex and requires more training: this

---

[1] in binary arithmetic the addition/multiplication algorithms start processing the LSBs in order to correctly propagate the intermediate carries.

[2] We used standard CrossEntropy loss, Adam optimizer with learning rate 0.0001 and betas = 0.9 and 0.98, and minibatch size = 128.

is due to the high non-linearity of this operation (more on this later) and to the higher length of the output (14 vs 8 tokens). On the workstation used (with a single Titan RTX GPU) training can be completed in just 8 and 46 minutes respectively. The accuracy on the validation set is very close to the training set, denoting almost perfect generalization on numbers never seen before. This is a somewhat surprising result, especially considering the limited size of the training data. No grokking[3] was observed (Nanda et al., 2023).

Unlike Nogueira et al. (2021) (see their Appendix B for a similar setup) we were able to learn addition with the native sinusoidal positional encoding. In Lee et al. (2023) additions can be effectively learnt by a simple LM, but to reach 100% accuracy the training set had to be balanced in terms of the operand magnitude (i.e., number of digits) and carry propagation. The effectiveness of our training procedure is probably due to the lower complexity determined by a small vocabulary and fixed-length representation. As to multiplication, Muffo et al. (2023) were not able to effectively learn two (decimal) digits multiplications, and Lee et al. (2023) had to provide extra intermediate steps in the prompt (denoted as *detailed scratchpad*). On the contrary our model effectively learnt multiplication of 7 binary digits operands: again the simplified setup may have been the key.

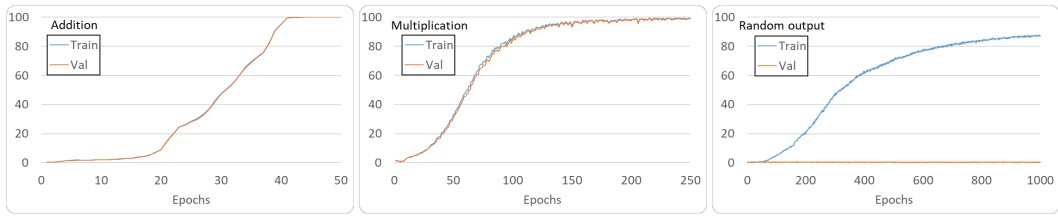

Figure 1: Sequence Accuracy. From the left: addition, multiplication and random output. Results are averaged over five runs.

## 4.2 CONTROL EXPERIMENT: RANDOM OUTPUT

If the output is randomly generated and therefore there is no relation with the input, the only possibility of learning the training set is memorizing the whole data. Figure 1(right) shows the training results: a much larger number of epochs (i.e., 1000) was necessary to reach a sequence accuracy of 87.8%, and, as expected, the validation accuracy did not increase over the epochs. The difficulty of memorizing the training set (many more epochs) is due to the high discontinuity of the input-output mapping. In fact, because of the random output generation, very similar input sequences can be associated to completely different outputs.

Therefore, even if we only consider the accuracy on the training set, this result shows that an exhaustive memorization of the input is much more complex for the LM than solving the addition and multiplication tasks. This leads us to assume that to efficiently solve the above computation tasks the LM has found a computational approach (or algorithm) to simplify the output prediction. Now the question is: what is the approach?

## 4.3 THE COMPUTATIONAL APPROACH

Let us consider two alternative approaches:

**Symbolic Manipulation (SM)**: a first idea is that the LM could learn the binary integer addition/multiplication algorithms used by an ALU inside a CPU (see Appendix B for a short reminder). Indeed, the addition algorithm is not complex and can be solved by using a 3-bit truth table (to sum each pair of corresponding bits with the carry-in) and iterative carry-out propagations. However, multiplication (by iterative additions) is much more complex and trickier to learn by using a symbolic manipulation approach. Furthermore, as shown by Lee et al. (2023) a simple LM can also learn complex operations such as the sine function or the square root, whose mathematical (and algorith-

---

[3]Grooking refers to the case where validation accuracy, much smaller than training accuracy at initial stages, suddenly increases after a certain number of epochs.

mical) decomposition is very complex since they require Taylor expansion and Newton method, respectively.

**Encoding-Regression-Decoding (ERD)**: if we consider the model architecture (Transformer) used for the LM and the underlying word embedding by vector representations, it is more likely that the LM solves the problem by decomposing it in the following three phases:

1. Encoding (token to value): mapping the input sequence (i.e., $a_0a_1a_2a_3a_4a_5a_6 \langle op \rangle$ $b_0b_1b_2b_3b_4b_5b_6$) to a suitable vector representation. In principle, two vectors $\boldsymbol{v}_A$ and $\boldsymbol{v}_B$ representing the values (or magnitudes) of $A$ and $B$ are enough.

2. Regression: learn the computation as a supervised regression problem in the vector space: $\boldsymbol{v}_R = regress(\boldsymbol{v}_A, \boldsymbol{v}_B)$. Actually this regression formulation is an oversimplification of the problem since in the next-token-prediction training the LM works incrementally. In Appendix C this discussion will be expanded.

3. Decoding (value to token): map the value vector $\boldsymbol{v}_R$ back to token representation (i.e., $r_0r_1...r_m$).

It is worth noting that the above Encoding and Decoding phases do not need to be mapped onto the Transformer encoder and decoder (more on this later). The experiments reported in Sections 4.4 and 4.5 support the ERD assumption. The capability of capturing number magnitudes by pre-trained embedders was also investigated by Wallace et al. (2019) who successfully trained a simple external regressor to compute the sum of two numbers starting from their embeddings. Other interesting studies on capturing numeracy with embedding were carried out by Naik et al. (2019) and Sundararaman et al. (2020).

## 4.4 INTERPOLATION VS EXTRAPOLATION

The random training/validation split performed for the experiments reported in Section 4.1 constitutes a somewhat simplified testbed to learn the two tasks. In fact, random split leads to a complete (even if sparse) coverage of the input space by both the training and validation set, where each example in the validation set has high chance to be close to a training set example, and interpolation is enough to fill the gaps.

Hereafter we considered two different criteria to isolate specific portion of the input space for the validation set, in order to better investigate extrapolation capabilities:

- $VS_t = \{(A, B)|(A, B) \in NN_{4096}((A^*, B^*))\}$

  where $NN_{4096}((A^*, B^*))$ is the set of 4096 pairs $(A, B)$ which are the nearest neighbors to a centroid $(A^*, B^*)$ according to the Hamming distance between the corresponding token representations (i.e., number of different tokens at corresponding positions). As centroid $(A^*, B^*)$ in the token space we used: $1010101 \langle op \rangle 0101010$.

- $VS_v = \{(A, B)|32 \leq A < 96 \quad \text{and} \quad 32 \leq B < 96\}$

  here the centroid is located in the middle of the value space (64, 64), so $VS_v$ is a squared region (of side 64) centered in the value space.

Both $VS_t$ and $VS_v$ isolate a contiguous data region of 4096 samples to be included in the validation set, but in the former the samples are close in the token representation space, while in latter are close in the value space. Being such contiguous portions of space excluded from the training set, we can expect a worse generalization. From the results (see Figure 2) we note that $VS_t$ is very marginally affecting LM training and generalization while $VS_v$ has a major impact: in fact, in the second case, for both addition and multiplication the final sequence accuracy is 4...6% points lower. This result strengthen the ERD hypothesis, since: (i) using $VS_v$ leads to the exclusion of a specific contiguous portion of value space during phase 2 and does not allow to properly train the regressor in this region; (ii) the encoding performed during phase 1 makes irrelevant the selection performed according to $VS_t$ because after encoding the corresponding data point remains scattered in the value space and the regressor can easily interpolate among them.

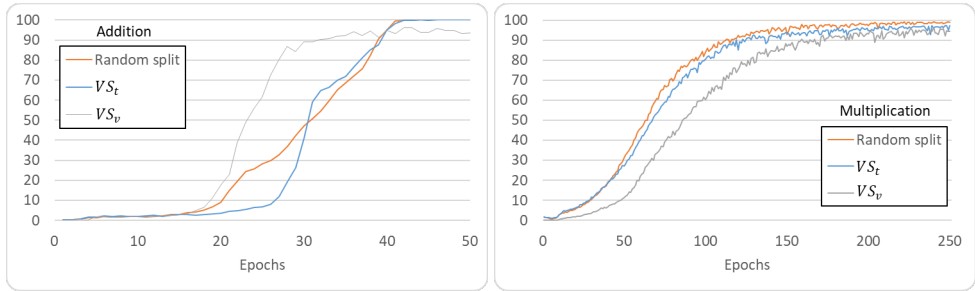

Figure 2: Sequence accuracy on Random, $VS_t$, and $VS_v$ validation subsets for addition (left) and multiplication (right). Results are averaged over five runs. $VS_t$ reaches 100% accuracy on additions (the same of Random split) and 97.5% accuracy on multiplication (just 1.4% less than random split); $VS_v$ reaches 93.7% on addition and 94.3% on multiplication (6.3% and 4.6% less than Random split, respectively).

### 4.5 LOOKING AT INTERNAL REPRESENTATIONS

Understanding internal representation (embeddings in the vector space) in a trained Transformer is not an easy task. However, in the specific setting considered we can gain some hints looking at the distances between the embedding of different data points (at different layers) and correlating them with the corresponding distances at input/output levels. Unlike probing, this approach does not require to rely on external models and well fit our aims.

Given an LM trained on addition (or multiplication) we consider the dataset S including the 128 input pairs where the two operands have identical value[4]: $S = \{(A, A) | 0 \leq A < 128\}$. At input level (*in*) we can compute two ordered sets of 8128 (128×127/2) distances each:

$$d_{in,t} = \{hdist(X,Y) | (X,X), (Y,Y) \in S, X < Y\}$$
$$d_{in,v} = \{|X - Y| \quad | (X,X), (Y,Y) \in S, X < Y\}$$

where $hdist(X,Y)$ is the Hamming distance between the token representation of $X$ and $Y$, and the subscript letters $t$ and $v$ denote token and value level, respectively.

At output level (*out*) we can compute the two corresponding sets of distances as:

$$d_{out,t} = \{hdist(P,Q) | (X,X), (Y,Y) \in S, X < Y\}$$
$$d_{out,v} = \{|P - Q| \quad | (X,X), (Y,Y) \in S, X < Y\}$$

where $(P = X + X$ and $Q = Y + Y)$ for addition, and $(P = X \times X$ and $Q = Y \times Y)$ for multiplication.

Finally, for each intermediate level of the Transformer encoder (*enc*) or decoder (*dec*) we can compute the Euclidean distances among the corresponding embedding vectors.

$$d_{enc_i} = \{\|enc_i(X,X) - enc_i(Y,Y)\| \quad | (X,X), (Y,Y) \in S, X < Y\}$$
$$d_{dec_i} = \{\|dec_i(X,X) - dec_i(Y,Y)\| \quad | (X,X), (Y,Y) \in S, X < Y\}$$

where $enc_i$ and $dec_i$ are the output vectors obtained by concatenating all the token embeddings (each of dimensionality 64) after the $i$-th encoder and decoder layer, respectively. For example $enc_i$ has dimensionality $960 = 64 \times 15$ where 15 is in the number of tokens in the encoder.

Even if the distances in the different sets have different ranges, we can use correlation to find out similarities. If two set of distances are correlated we can expect that the corresponding representations/embeddings are correlated as well. Since both Pearson and Spearman correlations (Schober et al., 2018) provided similar outputs, for simplicity in Figure 3 we report only Pearson correlations.

---

[4]since the input prompt contains two operands, we select only the cases with identical values ($A = B$) in order to easily determine the "magnitude" of the input, and thereafter compute meaningful distances.

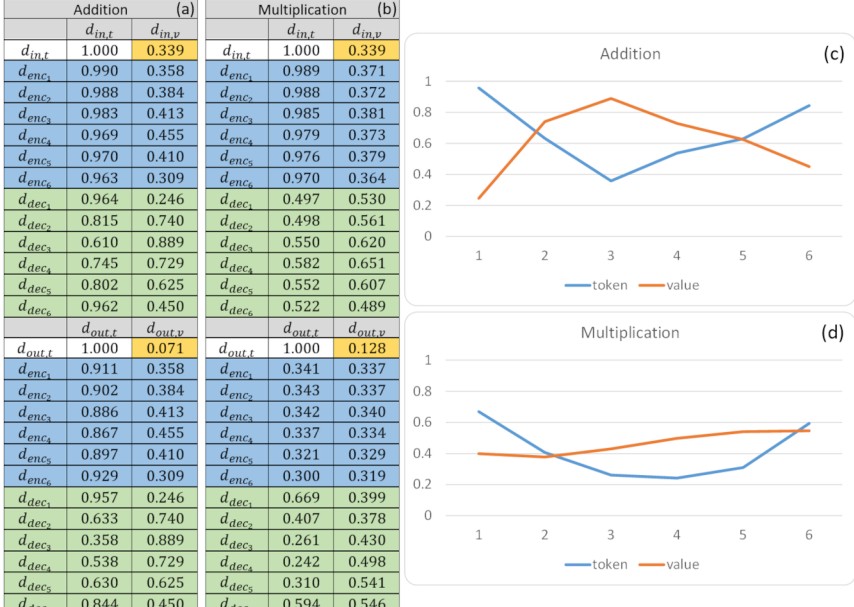

Figure 3: Pearson correlation between ordered sets of distances for addition (a) and multiplication (b). Each cell denotes the correlation between the two ordered set of distances specified in the corresponding row and column. Note that since for addition in this experiment the output value is always twice the input, the correlation values (blue and green cells) are the same for $d_{in,v}$ and $d_{out,v}$ block of values. Graphs (c) and (d) show the correlations of output distances $d_{out,t}$ (at token level - blue curves) and $d_{out,v}$ (at value level - orange curves) with the embedding distances $d_{dec_i}$ across the 6 decoder layers for addition and multiplication, respectively.

The yellow cells in the tables of Figure 3 confirm the low correlation between the token and value representation at both input and output level. The blue cells show that correlation remains quite similar across the encoder layers as if the encoder was not performing any significant computation (this is confirmed in Section 5 where by totally removing the encoder we achieve similar results). More interesting is the trend of correlations across the decoder layers (green cells). In particular, for the addition the token representation have high correlation with the first and last layers and low with central layers, while the value representation has opposite trend (see also Figure 3.c). These results support the ERD hypothesis and in particular that the initial and final layers in the decoder transform from token to value representation (and vice versa) while the central layers perform regression in the value space. In particular, at layer 3, the correlation at token level is minimum while the correlation at value level is maximum.

For multiplication the low-high-low trend at value level is less evident (Figure 3.d orange curve), probably because the quadratic dependence of the output from the input (at value level) does not allow to learn a simple regressor smoothly working in the whole vector space, and the mapping is performed by piecewise linear approximation in different space regions, which introduces discontinuities that makes global distances in the vector space unsuitable to quantify the representation similarity.

## 5 ABLATION STUDY

This section presents the results of an ablation study where the LM architecture was simplified, to understand what components are necessary to learn the addition/multiplication computation. Consistently with other studies, our results show that a decoder only architecture (Liu et al., 2018) can achieve similar results w.r.t. an encoder-decoder Transformer. A simplification of the architecture in terms of (i) reduction of dimensionality; (ii) reduction of number of heads; (iii) removal of fully connected layers is well tolerated, while positional embedding and attention layers are mandatory

Table 2: Epochs necessary to reach 95% accuracy on the validation set. A dash is used when 95% accuracy is not achieved in 1K epochs: in such case the accuracy reached is reported within brackets.

| Configuration | Addition | Multiplication |
|---|---|---|
| Full (see Table 1) | 39 | 137 |
| Decoder only | 60 | 426 |
| num_heads $h$=1 | 25 | 225 |
| Reduced dimensionality ($d_{model} = 32$) | 66 | 309 |
| No positional embedding | — (2.4%) | — (1.8%) |
| No attention layers | — (0.9%) | — (1.7%) |
| No fully connected layers | 56 | 398 |

for the LM in order to properly perform token to value transformation (and vice versa). Table 2 summarizes the results.

## 6  DISCUSSION AND CONCLUSIONS

In this paper we introduced a simplified setup to allow a light LM to learn binary addition and multiplication. The model easily learn the two tasks and generalize well on unseen data, proving that memorization of the training data is neither necessary nor efficient. The experiments on the interpolation/extrapolation capabilities and correlation of input-output representations with internal embedding suggest that the model solve the computational task as a supervised regression problem in the value space after an initial encoding from token to values, and a final decoding from output value to tokens. Under this hypothesis: (i) any task that can be solved by a neural network regressor can be solved by an LM as well, with the extra burden of end-to-end learning decoding/encoding steps; (ii) when looking at interpolation/extrapolation capabilities of an LM applied to a mathematical task, we should not concentrate on the input token representation but on the internal representation after encoding, keeping in mind the difficulties of a numerical regressor to work on region spaces not covered by the training set; (iii) on a more speculative side, we could guess that modern LLMs learn the number encoding/decoding once and reuse it across different numerical tasks whereas a specific regressor is learnt for each task.

Our ERD hypothesis could be questioned considering some recent findings from Lee et al. (2023) where providing in the prompt intermediate information (scratchpad) about the decomposition of arithmetic tasks improves the training efficiency and requires less examples. This could suggest that a symbolic manipulation approach is adopted to learn imitating step by step the proposed decomposition. However, in most of the cases their model was able to learn the same task (even if slowly) without scratchpad and/or with wrong scratchpads. As argued by the authors the higher efficiency is actually in terms of examples and not in terms of tokens since each scratchpad requires a large number of extra tokens, and we guess these could be used as extra features by the underlying regressor. Furthermore, scratchpad contribution is negligible for more complex operations such as sine and square root, but, unexpectedly, learning such complex operations was simpler than multiplication. This is not strange under the ERD hypothesis where a unary smooth operator like the sine can be learnt by a supervised regressor independently of the mathematical method used for its computation.

The algorithmic interpretation that Nanda et al. (2023) provided for modular addition, could also suggest that their LM discovered and efficient symbolic manipulation approach; however, as discusses in Section 2.3, it is more likely that a regressor was learnt to numerically approximate an efficient sparse Fourier decomposition, under regularization constraints favoring sparsity. Finally, the information flow described by Stolfo et al. (2023), points out that MLPs in the last layers are responsible of the numerical computation of the solution, which is compatible with the hypothesis of a multi-layer regressor.

Of course we are not claiming that all the capabilities of modern LLMs can be explained by regression, but regression is likely to be one of the internal tool that LLMs uses to predict next token when numbers come into play. To gain insights of other tools/mechanisms, one of the aim of our future research is designing simplified experiments/setups for tasks than cannot be easily mapped to regression problems such as chain of reasoning and logic deductions.

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

## A ADDITION INPUT-OUTPUT DISCONTINUITIES

In the example below a single bit modification in the input produces an 8 bit modification in the output for the addition task:

$$1000000 + 0111111 \rightarrow 11111110$$

$$1000000 + 1111111 \rightarrow 00000001$$

Generally, given an input/output pair, we consider the ($2^{14}$) variants obtained by perturbing (i.e., 0-1 swap) the input bits and counting the resulting changes in the output. These values, averaged over all possible input/output pairs (again $2^{14}$) and normalized by row are inserted in the cells of 3. So, for example the value in cell (row=2, column=3) means that in the 27.9% of the cases a perturbation of 2 (over 14) bits in the input leads to a change of 3 (over 8) bits in the output.

Table 3: Addition input-output discontinuities.

|    | 0 | 1 | 2 | 3 | 4 | 5 | 6 | 7 | 8 |
|----|------|------|------|------|------|------|------|------|------|
| 0  | 100.0% | 0.0% | 0.0% | 0.0% | 0.0% | 0.0% | 0.0% | 0.0% | 0.0% |
| 1  | 0.0% | 50.0% | 28.6% | 12.5% | 5.4% | 2.2% | 0.9% | 0.3% | 0.1% |
| 2  | 3.8% | 8.8% | 30.6% | 27.9% | 16.4% | 7.9% | 3.3% | 1.1% | 0.2% |
| 3  | 0.8% | 8.0% | 16.5% | 27.0% | 24.3% | 14.6% | 6.5% | 2.0% | 0.3% |
| 4  | 0.8% | 4.3% | 14.2% | 22.7% | 25.9% | 19.4% | 9.4% | 2.9% | 0.4% |
| 5  | 0.5% | 3.8% | 11.3% | 21.9% | 26.1% | 21.2% | 11.2% | 3.5% | 0.5% |
| 6  | 0.4% | 3.2% | 11.0% | 20.9% | 26.5% | 22.0% | 11.8% | 3.7% | 0.5% |
| 7  | 0.4% | 3.3% | 10.7% | 21.2% | 26.4% | 22.1% | 11.8% | 3.6% | 0.5% |
| 8  | 0.5% | 3.4% | 11.1% | 21.2% | 26.5% | 21.8% | 11.5% | 3.6% | 0.5% |
| 9  | 0.5% | 3.6% | 11.3% | 21.3% | 26.0% | 21.4% | 11.7% | 3.6% | 0.5% |
| 10 | 0.6% | 3.8% | 11.4% | 20.6% | 25.2% | 21.5% | 12.0% | 4.3% | 0.6% |
| 11 | 0.6% | 3.7% | 10.5% | 19.1% | 23.8% | 21.7% | 14.4% | 5.1% | 1.1% |
| 12 | 0.6% | 3.3% | 8.5% | 15.2% | 22.1% | 23.4% | 15.9% | 9.9% | 1.1% |
| 13 | 0.7% | 2.6% | 4.8% | 9.4% | 16.5% | 25.0% | 26.8% | 7.1% | 7.1% |
| 14 | 0.8% | 0.8% | 1.6% | 3.1% | 6.3% | 12.5% | 25.0% | 50.0% | 0.0% |

Input-output discontinuities, which are further amplified in case of multiplications, make it very unlikely to solve these tasks by smooth interpolation of the input representation.

## B BINARY ADDITION AND MULTIPLICATION

Binary addition can be executed by summing pairs of corresponding bits $a_i$ and $b_i$, starting from the LSBs ($a_0$ and $b_0$) and propagating carries. Let $c_{i-1}$ be the pending carry used to sum current bits [5], then a two-output 3-bit truth table (Table 4) can be used to generate the output bit $o_i$ and carry $c_i$ used when summing next pair of bits:

A simple approach to execute binary multiplication is through iterative binary sums. Each bit $b_i$ of the second operand is multiplied by the whole first operand, but this inner multiplication is straightforward since it results either in a sequence of 0 (if $b_i = 0$) or a copy of the first operand (if $b_i = 1$). This intermediate result is then shifted left and summed to the current output. An example is reported in Figure 4 below.

## C LEARNING A REGRESSOR UNDER PREDICT-NEXT–TOKEN TRAINING

In Section 4.3 we argued that an arithmetic computation task can be decomposed in three steps whose central one is learning a regressor in the value space: $\boldsymbol{v}_R = regress(\boldsymbol{v}_A, \boldsymbol{v}_B)$. If we consider the autoregressive working mode of a Transformer and its predict-next-token training, the regressor

---

[5]when summing the LSBs ($i = 0$), there is no pending carry, so $c_{-1} = 0$

Table 4: Two-output 3-bit truth table for binary addition.

| Inputs | | | Outputs | |
|---|---|---|---|---|
| $a_i$ | $b_i$ | $c_{i-1}$ | $o_i$ | $c_i$ |
| 0 | 0 | 0 | 0 | 0 |
| 0 | 0 | 1 | 1 | 0 |
| 0 | 1 | 0 | 1 | 0 |
| 0 | 1 | 1 | 0 | 1 |
| 1 | 0 | 0 | 1 | 0 |
| 1 | 0 | 1 | 0 | 1 |
| 1 | 1 | 0 | 0 | 1 |
| 1 | 1 | 1 | 1 | 1 |

```
    1100 (this is 12 in decimal)
  × 1001 (this is 9 in decimal)
  ------
    1100 (this is 1100×1)
   0000  (this is 1100×0, shifted left one position)
  0000   (this is 1100×0, shifted left two positions)
+ 1100   (this is 1100×1, shifted left three positions)
---------
 1101100 (this is 108 in binary)
```

Figure 4: Example of 4-digit binary multiplication. The sum can be performed incrementally with a two operands adder.

must be able to work incrementally given the output produced so far. In particular, we can formulate the problem as: $\boldsymbol{v}_{r_i} = regress(\boldsymbol{v}_A, \boldsymbol{v}_B, i, \boldsymbol{c}_{R_{i-1}})$ where:

- $\boldsymbol{v}_A = [\boldsymbol{v}_{a_0}\boldsymbol{v}_{a_1}...\boldsymbol{v}_{a_7}]$ and $\boldsymbol{v}_B = [\boldsymbol{v}_{b_0}\boldsymbol{v}_{b_1}...\boldsymbol{v}_{b_7}]$ are the value vectors of the two input operands, obtained as the concatenation of the value vectors of single tokens. Both are always fully available to the decoder. Note that, $\boldsymbol{v}_{a_i}$ and $\boldsymbol{v}_{b_i}$ are not the bits of the inputs, but correspond to their value vectors including also positional information.

- $i$ is the position of the token to be predicted (we can assume it is available through positional encoding).

- $\boldsymbol{c}_{R_{i-1}} = [\boldsymbol{c}_{r_0}\boldsymbol{c}_{r_1}...\boldsymbol{c}_{r_{i-1}}]$ is a value vector encoding the current context determined by the result produced so far (entering in the decoder from the bottom).

- $\boldsymbol{v}_{r_i}$ is the value vector of the $i$-th token.

In principle, the regressor could predict each $\boldsymbol{v}_{r_i}$ based on $\boldsymbol{v}_A$ and $\boldsymbol{v}_B$ alone, but we argue that the exploitation of the result produced so far $\boldsymbol{c}_{R_{i-1}}$ can lead to highest training efficiency. To this purpose is interesting to evaluate the impact of the output ordering (plain or reverse). In both the addition and multiplication the $i$-th token of the result only depends on the tokens of the inputs at positions $\leq i$ (see Appendix B). Therefore, if reverse order is adopted, as we assumed until now, $\boldsymbol{v}_{A_i} = [\boldsymbol{v}_{a_0}\boldsymbol{v}_{a_1}...\boldsymbol{v}_{a_i}]$, $\boldsymbol{v}_{B_i} = [\boldsymbol{v}_{b_0}\boldsymbol{v}_{b_1}...\boldsymbol{v}_{b_i}]$ and $\boldsymbol{c}_{R_{i-1}}$ are sufficient to predict $\boldsymbol{v}_{r_i}$. Viceversa, if the output computation starts with the MSBs the regressor cannot leverage the above iterative decomposition and need to learn the task as a global operation using whole vectors $\boldsymbol{v}_A$ and $\boldsymbol{v}_B$ with almost no support from the result produced so far.

In Figure 5 we note that with plain order both addition and multiplication require a much longer number of epochs to converge and the learning curve is less stable. Further experiments proved that, as expected, the order of the inputs (also reverse by default in this study) is irrelevant, since the LM can always access the whole input representations $\boldsymbol{v}_A$ and $\boldsymbol{v}_B$. The advantages of using the reverse order are pointed out in other recent studies (Nogueira et al., 2021; Lee et al., 2023). In particular, Lee et al. (2023) reported a significant improvement w.r.t. plain order.

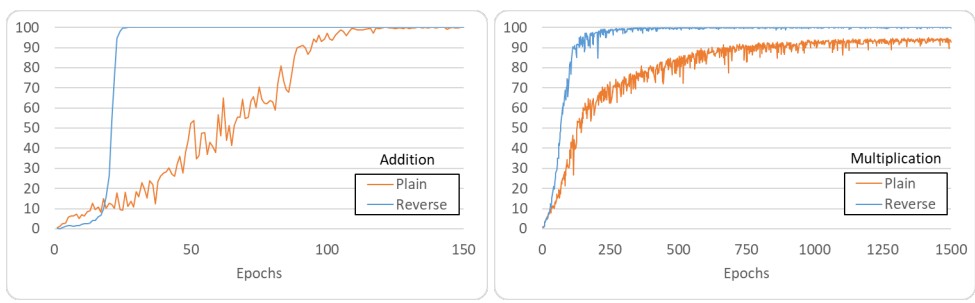

Figure 5: Sequence accuracy on validation set for reverse (default in this work) and plain order of the input and output representations. From left to right: addition and multiplication.

