# OpenReview forum: "Arithmetic with Language Models: from Memorization to Computation"
_ICLR.cc/2024/Conference — Submitted to ICLR 2024_

### Official Review · Reviewer_2eeE · 2023-11-01

**Soundness:** 2 fair
**Presentation:** 2 fair
**Contribution:** 2 fair
**Rating:** 5
**Confidence:** 4

**Summary:**

A transformer encoder-decoder is trained to perform binary addition and multiplication, and (somewhat surprisingly) it learns multiplication well. Two hypotheses are investigated for how it multiplies: either by manipulating symbols or by using activations to estimate the magnitude of the multiplicands and product.

**Strengths:**

The two hypotheses for how the transformer might do multiplication make sense and are interesting. The experiments in 4.5 seem to point to the ERD hypothesis.

**Weaknesses:**

It's a little bit mysterious why previous authors have observed that it's difficult for a transformer to learn multiplication, but here it learned it perfectly.

Since the paper is arguing that the ERD hypothesis is correct, I would have appreciated a more detailed explication of what the transformer might be doing -- ideally, an explicit construction of a transformer that multiplies, but if that's not possible, at least some kind of sketch.

Section 4.6 was not very clear to me. Previously (page 5) $v_A$, $v_B$, and $v_R$ were stated to be vector representations of the values (as opposed to the binary represntations) of $A$ and $B$. But in 4.6, $v_A$ has become the concatenation of the vectors of $a_0, a_1$, etc., which are the bits of $A$? This seems more like symbolic manipulation to me.

The ablation study in section 5 shows that the position embedding is necessary, but this has to be so, because otherwise the encoder would know nothing about the order of the digits.

**Questions:**

Could you explain Section 4.6 a bit more?

---

> ### Author Response · Authors · 2023-11-21
>
> Hereafter we reply to your specific Questions. Please also have a look at the global responses where we summarize our general comments.
>
> Q1: Section 4.6 (now moved to Appendix for lack of space) provides more details on the concept introduces in 4.3 considering the autoregressive nature of the LM. In this case v_A and v_B are the composition of value vectors (each deriving from a single token = bit) but not the composition of original bits. So the computation still take place in the embedding space in vectorial form (without explicit bit manipulation). Some further comments have been added to the new version of this section.
>
> Concerning the internal implementation of multiplication, we believe that a good parallel is considering an MLP with two (numerical) inputs for the operands, one output neuron for the result and some intermediate levels. You can train such simple model to learn multiplication as a regression problem; in case of Relu activation function, it is likely that the solution is a piecewise linear approximation of the non-linear surface.

---

### Official Review · Reviewer_Bkzb · 2023-11-01

**Soundness:** 3 good
**Presentation:** 3 good
**Contribution:** 3 good
**Rating:** 6
**Confidence:** 2

**Summary:**

This work investigates how arithmetic operations like addition and multiplications happen inside a Language Model (LM). They train a small language model using few tokens to show that language model works as encoding-regression-decoding machine where the computation take place in value space. To show this claim they train LM with different configurations of training datasets by first showing the language model doesn't memorize the input and output and then showing how excluding certain regions from training set leads to degradation in performance of language model.

**Strengths:**

- Clean experiments with each section supporting the claim they want to make.
- Explains how arithmetic operations works inside a language models and showing they act as encoding-regression-decoding machines.

**Weaknesses:**

- Explores it for very simple LM not sure if the results generalize to large LLMs. Showing results on different open-source LLMs might be helpful to make the claim stronger.
- Lack of novel insights after reading the paper I am like okay they act as encoding-regression-decoding machines but I don't know how to use this information to build models better at doing arithmetic computation.

**Questions:**

Asked in the weakness section.

---

> ### Author Response · Authors · 2023-11-21
>
> Hereafter we reply to your specific Questions. Please also have a look at the global responses where we summarize our general comments.
>
> Both points are addressed in global responses.

---

### Official Review · Reviewer_exQX · 2023-11-01

**Soundness:** 2 fair
**Presentation:** 1 poor
**Contribution:** 2 fair
**Rating:** 3
**Confidence:** 3

**Summary:**

This paper investigates language models' ability to perform arithmetic computations.
It focuses on language models trained on a single arithmetic operation—either addition or multiplication—which operates on binary natural numbers between 0 and 127.
These numbers are zero padded in their most significant digits (so they always have the same length when input to the models), and are reversed (going from least significant to most significant digit).
The paper proposes a Encoding-Regression-Decoding hypothesis, arguing models encode input digits, perform a regression task in its hidden states space, and then decode the output digits.
The paper then runs a number of experiments:
* **Performance in distribution.** In the first experiment, the models are simply trained and evaluated on i.i.d train-validation split. The models reach ~100% accuracy in both train and validation.
* **Memorisation baseline.** In the second experiment, models are trained on a dataset for which outputs are randomly sampled. In this setting, the model takes much longer to reach a reasonable training performance, and never passes chance validation performance.
* **Performance out of distribution.** In the third experiment, models are trained on two train--validation splits designed to evaluate o.o.d. performance. Two splitting strategies are proposed: (i) one creates o.o.d. splits based on the hamming distance between the strings (e.g., 01101000) representing the input digits, and (ii) the second creates o.o.d. splits by putting all inputs representing numbers from 32 to 64 in a validation set. In these experiments, the authors find that models have a harder time generalising in the second setting, and take this as evidence of the Encoding-Regression-Decoding hypothesis.
* **Embedding Distance and Correlations.** The authors also: (i) get the hidden state of their model when input the same value twice (e.g., when input $A+A$), (ii) compute the distance between each pair of hidden states representing $A+A$ with $A \in [0, 127]$, (iii) compute the (Hamming and value) distance between each pair of  inputs $A \in [0, 127]$, (iv) evaluate the correlation between hidden state and input distances. In this task, the authors find that hidden state distances correlate more strongly with Hamming distances in early and final layers, but with numerical distances in middle layer. This is taken as evidence for the Encoding-Regression-Decoding hypothesis.
* **Performance with reverse vs normal strings.** The authors compare model performance when predicting reversed or normal digits. They show learning arithmetic tasks with reversed digit strings is easier, which they take as evidence that language models compute outputs iteratively one bit at a time.

**Strengths:**

This paper tackles an important and interesting question.

The simplified setting which is analysed here allows the authors to isolate training/optimisation issues (to some extent) and analyse the strategy used by the models to perform arithmetic tasks.

The experiments in the paper are clearly written, and give important insights about how language models implement arithmetic tasks.

**Weaknesses:**

The main weakness of this paper, in my opinion, is that it does not engage with the model interpretability literature (neither in “mechanistic interpretability” or “probing”).
* It cites a single probing paper on probing numeracy in embeddings, which is a highly relevant topic here, but many more exist (e.g., Naik et al. 2019, Sundararaman et al. 2020).
* It cites no work on probing, many of which have discussed techniques similar to the presented here. E.g., manipulating datasets on which a model is trained/evaluated to analyse its computational strategy (e.g., Linzen et al. 2016, Warstadt et al. 2020). E.g.2, computing distances in embedding space which are representative of other notions of distances (e.g., Hewitt and Manning. 2019, White et al. 2021,  Limisiewicz and Mareček. 2021)
* Many issues have been pointed out with this type of non-causal probing analyses (e.g., Elazar et al. 2021, Ravfogel et al. 2021, Lasri et al. 2022) which would be interesting to acknowledge.
* It cites no related mechanistic interpretation work either. Hanna et al. (2023), in particular, study how GPT models implement a greater than operation, which is quite related to the research question investigated here.

Related to the issue above, the set of experiments already in this paper are a nice start on the quest to understand how arithmetic operations are implemented in language models. But this paper’s contributions could be much stronger if the authors engaged with the literature above and applied some of the more recent probing/interpretability techniques in their analyses. As is, I do not think this paper’s contributions are strong enough to warrant publication in ICLR.


## References

* Naik et al. 2019. Exploring Numeracy in Word Embeddings. In: ACL.
* Sundararaman et al. 2020. Methods for Numeracy-Preserving Word Embeddings. In: EMNLP.
* Linzen et al. 2016. Assessing the Ability of LSTMs to Learn Syntax-Sensitive Dependencies. In: TACL
* Warstadt et al. 2020. Learning Which Features Matter: RoBERTa Acquires a Preference for Linguistic Generalizations (Eventually). In: EMNLP.
* Hewitt and Manning. 2019. A Structural Probe for Finding Syntax in Word Representations. In: ACL.
* White et al. 2021. A Non-Linear Structural Probe. In: NAACL.
* Limisiewicz and Mareček. 2021. Introducing Orthogonal Constraint in Structural Probes. In: ACL.
* Elazar et al. 2021. Amnesic Probing: Behavioral Explanation with Amnesic Counterfactuals . In: TACL
* Lasri et al. 2022. Probing for the Usage of Grammatical Number. In: ACL.
* Ravfogel et al. 2021. Counterfactual Interventions Reveal the Causal Effect of Relative Clause Representations on Agreement Prediction. In: CoNLL.
* Hanna et al. 2023. How does GPT-2 compute greater-than?: Interpreting mathematical abilities in a pre-trained language model. In: NeurIPS

**Questions:**

I thought the paper was relatively clear and I have no specific questions. I am not sure if any specific response from the authors would change my opinion regarding this paper, but maybe they would wish to address the lack of engagement with the literature in their response, or why the referred papers are in their opinion perhaps not relevant in this case?

## Minor Presentation Issues

I thought section 4.5 was confusing at a first read because you use the same notation ($A$ and $B$) there as in previous sections, but you refer to different inputs in this section, as opposed to the two values in a single input. This could likely be made clearer to the reader.

Plots are not currently readable in black and white.

Citation Wei et al. (Chain of thought) is currently duplicated.

---

> ### Author Response · Authors · 2023-11-21
>
> Hereafter we reply to your specific Questions. Please also have a look at the global responses where we summarize our general comments.
>
> Q1: Thank you very much for the detailed review and the relevant references provided. As explained in the general comments a new version has been uploaded where the work is better framed into the interpretability literature.
>
> Moreover, we fixed the minor presentation issues (except the plot colors that can be easily adjusted in the final version).

---

> ### Comment · Reviewer_exQX · 2023-11-22
>
> I thank the authors for their response and for adding an additional section where they discuss related work.
> I am keeping my score, however, since I still think this paper could engage with prior work more thoroughly. That would likely require a deeper discussion of prior work, and adopting some of those techniques to this work (or motivating why new techniques are required).
> As mentioned in my review, I still believe what this paper does is a combination of:
> * Behavioural probing: manipulating datasets on which a model is trained/evaluated to analyse its computational strategy (e.g., Linzen et al. 2016, Warstadt et al. 2020).
> * Traditional probing: computing distances in embedding space which are representative of other notions of distances (e.g., Hewitt and Manning. 2019, White et al. 2021, Limisiewicz and Mareček. 2021).
>
> It's not because the authors use no probe here, that their embedding distance correlational analysis is less susceptible (than, e.g., Hewitt and Manning's approach) to the criticism put forward by causal probing papers. (Or maybe there is some reason why the criticism does not apply to this work, but I did not find that argument in the paper.)
> I would recommend the authors try to apply some causal probing techniques here, such as, e.g., Geiger et al. (2021, 2023).
>
> Related to the new sentence "Mechanistic interpretability is still more ambitious, since it is aimed at reverse engineering the algorithm that a model uses to solve a task". I don't think this is true. As far as I know, there is no difference between the goals of "causal probing" and "mechanistic interpretability". I think these are just two names for the same thing (which were developed in parallel by different sub-communities doing interpretability of neural models).
>
> * Geiger et al. 2021. Causal abstractions of neural networks
> * Geiger et al. 2023. Finding alignments between interpretable causal variables and distributed neural representations

---

### Official Review · Reviewer_bFAd · 2023-11-06

**Soundness:** 2 fair
**Presentation:** 2 fair
**Contribution:** 1 poor
**Rating:** 3
**Confidence:** 3

**Summary:**

The paper delves into the area of mathematical reasoning within Large Language Models (LLMs), with a specific emphasis on Transformer-based models and their handling of arithmetic tasks. The core investigation revolves around understanding the computational mechanisms and processes employed by LLMs when performing arithmetic operations. The authors ran their investigations on toy encoder-decoder models (likely similar to the T5 models).

**Strengths:**

* Exploring the limitations of Large Language Models (LLMs) and the transferability of findings, especially in arithmetic tasks, is both relevant and crucial for the community at this time.

**Weaknesses:**

* The experimental setup is quite basic, and it's unclear how these findings apply to current Large Language Models. The research primarily focuses on binary addition and multiplication using a simplistic model, which might not be representative of more complex, real-world scenarios.

* The paper could benefit from clearer writing. Specifically, the abstract and introduction lack clarity regarding the nature of the investigation. It's not immediately apparent what the central findings are, the experiments conducted to arrive at these conclusions, and how this differs from existing knowledge in the field.

* There is a noticeable absence of some relevant literature. The authors should consider reviewing "A Mechanistic Interpretation of Arithmetic Reasoning in Language Models using Causal Mediation Analysis (Stolfo et al. 2023)," as well as other works cited in that paper, which seem pertinent to this study.

* Prior research indicates that certain types of positional encoding, such as Sinusoidal Positional Embedding—a fixed form of positional encoding—struggle with extrapolation to unseen positions. The use of this method in the toy models might result in outcomes more attributable to positional embedding issues rather than the arithmetic aspects under investigation.

**Questions:**

* Considering the experimentation was carried out in a very constrained setting, how do the findings of this work extend to broader applications or more complex scenarios? Specifically, what broader implications or learnings can be derived from the study's outcomes?

* What advantages does investigating encoder-decoder models offer in the context of this research, as opposed to exclusively examining a GPT-like model (i.e., decoder-only)? Additionally, how do you anticipate the results or observations might differ if the study were to shift its focus to decoder-only models?

---

> ### Author Response · Authors · 2023-11-21
>
> Hereafter we reply to your specific Questions. Please also have a look at the global responses where we summarize our general comments.
>
> Q1: Addressed in global responses.
>
> Q2: This was partially addressed in the paper by noting in Section 5 that “…encoder was not performing any significant computation (this is confirmed in Section 5 where by totally removing the encoder we achieve similar results)”. So we do not expect relevant changes when switching to GPT-like models.

---

### Author Response · Authors · 2023-11-21

Thanks to all the reviewers for the useful and constructive comments.

1. We are glad that the approach (and experiment design) was appreciated by 3 of 4 reviewers:
(rev2) “The simplified setting which is analysed here allows the authors to isolate training/optimisation issues (to some extent) and analyse the strategy used by the models to perform arithmetic tasks. The experiments in the paper are clearly written, and give important insights about how language models implement arithmetic tasks”.
(rev3) “Clean experiments with each section supporting the claim they want to make. Explains how arithmetic operations works inside a language models and showing they act as encoding-regression-decoding machines”.
(rev4) “The two hypotheses for how the transformer might do multiplication make sense and are interesting. The experiments in 4.5 seem to point to the ERD hypothesis”.
2.	We fully agree with rev2 (also pointed out by rev1) that the main weakness of the paper is (hopefully “was”) that “it does not engage with the model interpretability literature (neither in “mechanistic interpretability” or “probing”)”. Therefore, a new version was uploaded where related interpretability literature is now briefly discussed in the new subsection 2.2. Moreover, some of the suggested references (from rev1 and rev2) and some papers there cited allowed us to find further evidences to support our ERD hypotheses (discusses in the new subsection 2.2 and 6). The new material included led to an excessive length so we moved Section 4.6 to Appendix.
3.	A weakness pointed out by rev1 (and partially from rev4) is that “The experimental setup is quite basic, and it's unclear how these findings apply to current Large Language Models”. We think that the choice of simplified settings is necessary to deal with the complexity of interpretation (this is typical in most of the works in mechanistic interpretability, see subsection 2.3). Furthermore it is certainly true that some capabilities emerged only with scale, but not the vice versa. In other words, if a certain tool (e.g. internal regression) can be exploited by a small LM to solve a problem we believe that an LLM can easily discover and use it if needed.
4.	Finally, concerning the “broader implications and learning that can be derived from the paper”  (pointed out by rev1  and rev3) we think that an important aspect was mentioned in the conclusions: “when looking at interpolation/extrapolation capabilities of an LM applied to a mathematical task, we should not concentrate on the input token representation but on the internal representation after encoding, keeping in mind the difficulties of a numerical regressor to work on region spaces not covered by the training set;”. We believe this can help explaining limitations and designing better training strategies/datasets.

Specific replies to the revs Questions are provided below (after each review).
In the updated manuscript the changes are highlighted in red.

---

### Meta-Review · Area_Chair_J9i8 · 2023-12-13

**Metareview:**

This paper explores the capability of language models to perform arithmetic computations. The study focuses on language models that have been trained on a single arithmetic operation, either addition or multiplication, and operate on binary natural numbers between 0 and 127. The investigation was conducted using simple encoder-decoder models. This research addresses an important and intriguing question and provides valuable insights into how language models execute arithmetic tasks. However, one significant weakness pointed out by several reviewers is that the paper does not engage with the literature on model interpretability. Incorporating relevant literature and applying some of the latest probing/interpretability techniques would significantly enhance the quality of the paper. In addition, the paper uses a very simple language model, and it may be beneficial to utilize different open-source LLMs to strengthen the claims made in the study.

**Justification For Why Not Higher Score:**

One significant weakness pointed out by several reviewers is that the paper does not engage with the literature on model interpretability. Incorporating relevant literature and applying some of the latest probing/interpretability techniques would significantly enhance the quality of the paper. In addition, the paper uses a very simple language model, and it may be beneficial to utilize different open-source LLMs to strengthen the claims made in the study.

**Justification For Why Not Lower Score:**

N/A

---

### Decision · Program_Chairs · 2024-01-16

Reject